# Functionally Validating Evolutionary Conserved Risk Genes for Parkinson’s Disease in *Drosophila melanogaster*

**DOI:** 10.3390/insects14020168

**Published:** 2023-02-09

**Authors:** Amalie Elton Baisgaard, Kristina Magaard Koldby, Torsten Nygård Kristensen, Mette Nyegaard, Palle Duun Rohde

**Affiliations:** 1Department of Biomedicine, Aarhus University, 8000 Aarhus C, Denmark; 2Department of Health Science and Technology, Aalborg University, 9220 Aalborg, Denmark; 3Department of Chemistry and Bioscience, Aalborg University, 9220 Aalborg, Denmark

**Keywords:** model organism, neurodegenerative disease, RING assay, climbing ability

## Abstract

**Simple Summary:**

The neurodegenerative disorder Parkinson’s disease (PD) has a multifactorial etiology, and recent large-scale genetic studies have identified more than one hundred regions in the human genome that display statistical association with the risk of developing PD. To understand the disease mechanism better, biological validation of the identified genomic regions is warranted. However, in most studies such validation is absent. The aim of the study was to identify PD-genes that are evolutionary conserved between humans and the vinegar fly, and phenotypically investigate the effect of gene expression knockdown of those genes in the fly model. We identified eleven PD-genes that display strong evolutionary conservatism, and we successfully reduced gene expression in nine of the knockdown lines. A phenotype that previously has been used to investigate PD in a fly model is the flies’ innate escape response. We found that gene expression knockdown resulted in a disrupted escape response in eight of the nine knockdown lines. These results provide additional support for potential involvement of these genes in the disease pathology underlying PD.

**Abstract:**

Parkinson’s disease (PD) is a heterogeneous and complex neurodegenerative disorder and large-scale genetic studies have identified >130 genes associated with PD. Although genomic studies have been decisive for our understanding of the genetic contributions underlying PD, these associations remain as statistical associations. Lack of functional validation limits the biological interpretation; however, it is labour extensive, expensive, and time consuming. Therefore, the ideal biological system for functionally validating genetic findings must be simple. The study aim was to assess systematically evolutionary conserved PD-associated genes using *Drosophila melanogaster*. From a literature review, a total of 136 genes have found to be associated with PD in GWAS studies, of which 11 are strongly evolutionary conserved between *Homo sapiens* and *D. melanogaster*. By ubiquitous gene expression knockdown of the PD-genes in *D. melanogaster*, the flies’ escape response was investigated by assessing their negative geotaxis response, a phenotype that has previously been used to investigate PD in *D. melanogaster*. Gene expression knockdown was successful in 9/11 lines, and phenotypic consequences were observed in 8/9 lines. The results provide evidence that genetically modifying expression levels of PD genes in *D. melanogaster* caused reduced climbing ability of the flies, potentially supporting their role in dysfunctional locomotion, a hallmark of PD.

## 1. Introduction

Parkinson’s disease (PD) is the second most common neurodegenerative disorder worldwide [1], with a prevalence of approximately 2% of those >65 years of age [2]. PD is a multifactorial and heterogenic disorder and is characterised by patients displaying either motor or non-motor symptoms, or a combination. The motor symptoms typically include tremor, rigidity, slowness, and balance problems, whereas the non-motor symptoms include cognitive impairment, sleep disorders, and depression [1,3]. The symptoms occur due to loss of dopaminergic neurones primarily in the substantia nigra pars compacta, which project to the striatum; however, when the symptoms first appear, approximately 50% of the dopaminergic neurones have already been lost [3]. One of the major histopathological hallmarks of PD is the presence of aggregated α-synuclein protein, encoded by the *SNCA* gene, which can be identified as proteinaceous inclusions in the neurones, known as Lewy bodies [4,5].

Between 5 and 10% of all PD cases can be attributed to monogenic causes. For example, mutations within the *Parkin* (*PRKN*), *PTEN Induced kinase 1* (*PINK1*), *DJ-1* (*PARK7*), and *NUS1* genes are linked to early-onset autosomal recessive PD [6,7], whereas pathogenic mutations in the *Synuclein alpha* (*SNCA*), *Leucine rich repeat kinase 2* (*LRRK2*), and *Vacuolar Protein sorting-associated Protein 35* (*VPS35*) genes have been linked to autosomal dominant PD [7,8]. The remaining sporadic cases are characterised as being a multifactorial form of PD that is influenced by both environmental factors and many common genetic variants with small risk effects [9], for example, genetic variation within *cyclin-G-associated kinase* (*GAK*) gene has been associated with increased SNCA expression and increased PD risk [10,11].

Advancements in sequencing technologies and the establishment of large disease-specific cohorts and biobanks have enabled studies investigating the genetic aetiology underlying sporadic PD. At present, genome-wide association studies (GWAS) have identified >130 common single nucleotide polymorphisms (SNPs) associated with the risk of developing PD [9,12,13,14]. Despite the great success in identifying novel risk variants for sporadic PD, the genetic variants together only account for a minor proportion of the heritable risk underlying the disease. GWAS mainly provide associations between a disease outcome and polymorphic sites in the genome, which to some degree limits the biological interpretation. Therefore, functional assessment and validation of the identified candidate loci may help to uncover the biological aetiology of the disease. As large and well-powered GWAS often result in long lists of disease-associated loci, the biological system needed to validate those findings must be simple and efficient, and one such candidate is the vinegar fly, *Drosophila melanogaster*.

*D. melanogaster* has previously been used to study the human pathobiology of PD [6,15,16,17]. *D. melanogaster* as a model organism per se is popular as the flies are relatively easy and cheap to maintain in the laboratory, have a short generation time, no ethical restrictions, produce many offspring, and a large repertoire of genetic tools for gene manipulation exists [18,19,20]. In the context of PD, loss of *NUS1* expression (*NUS1* has been shown to be enriched for rare nonsynonymous mutations in PD cases [6]) in *D. melanogaster*, and has been shown to reduce climbing abilities of the flies (which is a hallmark of *D. melanogaster* ageing [21,22]), reduce the level of dopamine, and reduce the number of dopaminergic neurones within the flies [6]. Furthermore, it has been found that loss-of-function of the familial PD-associated genes *PINK1* and *PRKN* leads to learning and memory abnormalities and weakening of circadian rhythms, due to electrophysiological changes in clock neurones in flies [17]. Further, downregulation of *auxilin*, the *D. melanogaster* ortholog to the human *GAK* gene, leads to progressive loss of climbing ability in the flies compared with controls [23]. The climbing ability of the flies was assessed using the Rapid Iterative Negative Geotaxis (RING) assay which assesses the flies’ innate escape response [21,23]. The escape response is initiated by tapping the flies to the bottom of a vial, and afterwards assessing their distance moved within a specific timeframe [21,23]. Moreover, it has been found that transgenic flies producing normal human α-synuclein and flies carrying the A30P or A53T mutations of α-synuclein, both of which are linked to familial PD, all show an age-dependent decline in climbing ability from 23 to 45 days of age, compared with controls [24]. Collectively, these examples illustrate the applicability of *D. melanogaster* to study familial PD associated genes; however, using *D. melanogaster* to systematically study small effect PD GWAS loci has to our knowledge not been performed.

The aim of the current study was to use *D. melanogaster* as model system to assess a subset of highly conserved GWAS candidate genes for sporadic PD. By systematically reviewing the most recent large-scale human genetic studies of PD, a total of 136 unique genes statistically associated with PD were identified. Among the 136 potential candidate genes for sporadic PD, 11 genes found to be highly evolutionary conserved between *H. sapiens* and *D. melanogaster* were selected for inclusion in the current study. By ubiquitous gene expression knockdown of the selected PD risk genes, the flies’ innate escape response was investigated by assessing their negative geotaxis response using the RING assay [21], an assay that previously has successfully been used to investigate PD-like phenotypes using *D. melanogaster* as a model system [23,25,26,27].

## 2. Materials and Methods

### 2.1. Identification of PD Risk Genes

From the four most resent large genome-wide association studies (GWAS) of sporadic PD, we identified those genes that harboured at least one genome-wide significant (i.e., *p*-value < $5\times 10^{-5}$) single nucleotide polymorphism (SNP).

The GWAS studies included were: Chang et al. (2017), studying 26,035 PD cases and >400,000 controls [13]; Foo et al. (2019), investigating 6724 PD cases among 31,757 samples [14]; Nalls et al. (2019), in which 56,306 PD cases (or proxy cases, i.e., individuals who do not have PD but have a first-degree relative that does) and 1,417,791 controls were used in a large genetic meta-analysis [9]; and finally a longitudinal study by Iwaki et al. (2019) following 4093 PD patients for almost four years, resulting in a total of 22,307 phenotypic observations [12].

Thus, from those four large GWAS of PD an aggregated list of genes associated with PD was constructed. This combined list of previously associated PD risk genes was the starting point of the current study.

### 2.2. Drosophila Knockdown Lines and Maintenance

*Drosophila* orthologs of all identified PD associated genes were identified using the *Drosophila* RNAi Screening Centre (DRSC) integrative ortholog prediction tool (DIOPT) [28,29]. Only highly conserved genes were selected for functional assessment. This selection was based on two criteria: (1) the DIOPT orthology score should be >12 (i.e., more than 12 of the 15 available orthology sources agreeing on orthology [per April 2021]), and (2) a homozygous viable RNA interference (RNAi) line should be available. Information on all 136 human PD risk genes, their predicted *Drosophila* orthologs, including information on expression patterns (based on information from FlyBase [30,31]), is collected in Appendix A.

Fly stocks were maintained for two generations prior to the experiment on standard *Drosophila* Leeds medium, see Kristensen et al. (2016) [32], in a 23 °C climate room, 50% relative humidity, and a 12:12 h light/dark cycle. A total of 11 *UAS*-RNAi lines (Appendix A) were obtained from Vienna *Drosophila* Stock Centre (VDRC) [33], including the isogenic host line for the RNAi library (*w*_1118_, ID 60000). Ubiquitous gene expression knockdown was obtained by crossing females from the *UAS* lines (and the *w*_1118_ background) to males from the *Act5C*-Gal4 line (Bloomington *Drosophila* Stock Centre #4414). Under light CO_2_ anaesthesia, the sex of flies was determined and five biological replicates for each line containing fifteen individuals (ten females and five males) was established and separated into five distinct vials. For 5 consecutive days flies laid eggs in vials and every 12 h the adult flies were transferred to new vials (to avoid over-crowding). In the subsequent generation, ten virgin *UAS* females were collected every sixth hour and pooled with five *Act5C*-Gal4 males. For five days, the *UAS* and *Act5C*-Gal4 flies were transferred to new vials, allowing reproduction in several vials. This procedure was performed for all the 11 PD risk genes and the *w*_1118_ control line. Male offspring carrying the *UAS*-Gal4 construct were collected as they hatched and were used in the RING behavioural assay, as described below.

### 2.3. RING-Assay

Negative geotaxis was assessed with the RING-assay [21] (Appendix A). During the establishment of each knockdown line and the *w*_1118_-Gal4 control line, five biological replicates were established (see Section 2.2). From each biological replicate, 3 technical replicates were generated by sampling up to 15 males divided into three separate vials (i.e., up to 75 males per knockdown line). Only male flies were tested due to logistical reasons. At the age of 5, 10 and 15 days ± 12 h, the flies were transferred to empty vials, which were placed in an open-faced box (which constitutes the RING-apparatus) containing 10 vials, secured with a lid, and allowed to recover for 1 min. The RING apparatus was then tapped down onto the table three times and a picture was taken after 3 s of the last knockdown to capture the flies’ position in the vials for subsequent assessment of the flies’ moveability. The picture was taken with an iPhone 12 Pro (Apple Inc., Cupertino, CA, USA) with a countdown timer set to 3 s placed at a 30 cm distance from the RING apparatus. A total of five pictures were captured for each loading of the RING apparatus, with a 30 s break in between knockdowns. The position of the flies after knockdown was manually measured using ImageJ software (version 1.53a, Research Services Branch, National Institute of Mental Health, Bethesda, MD, USA). After the RING assay, the flies were transferred back to food vials and were tested in the RING assay again 5 days later. This was repeated twice, allowing testing of flies when they were 5, 10 and 15 days of age, as mentioned above. After the last RING-assay (i.e., at age 15 days) flies were snap-frozen in liquid nitrogen in Eppendorf-tubes and stored at −80 °C until further use.

### 2.4. RNA-Sequencing and Data Analysis

To quantify the level of gene expression knockdown of the 11 target genes, we performed RNA sequencing of pools of approximately 20 *D. melanogaster* heads from each knockdown line and the control line (collected after the last RING assay). The flies were decapitated when frozen and kept on dry ice to avoid thawing. RNA from fly heads was purified using the E.Z.N.A^®^ Total RNA Kit I (Omega, Bio-Tek, Norcross, GA, USA) according to manufacturer’s protocol. RNA sequencing was performed by BGI Global Genomic Services, Copenhagen, Denmark, with 100 bp paired end sequencing on a DNBSEQ platform (Appendix A).

RNA sequencing data were processed using the Partek^®^ flow^®^ software, v10.0* (Partek Inc., Chesterfield, MO, USA). Raw sequencing reads were aligned to the *Drosophila melanogaster* Reference Genome version BDGP6 [30] using STAR [34]. For quality control, reads with a mapping quality <20 and duplicate sequencing reads were removed. Further, for each sample, reads were excluded if mapping was within +/−100 bp of the corresponding RNAi construct.

Annotation was performed to RefSeq genes in the reference genome (version BDGP6) [30,35], and a count matrix was generated for genes in which (1) the sum of reads across all samples was ≥10 and (2) the maximum number of reads across samples was ≤10. Finally, normalisation between samples was performed using trimmed mean of M-values (TMM) normalisation [36].

For each knockdown line, the TMM of the RNAi target gene was compared with the TMM of the respective target gene within the common control (*w*_1118_-Act5C-Gal4). The change in gene expression was quantified as:(1)$$\mathrm{fold}-\mathrm{change}=log_{2}\left(\frac{TMM_{UAS-GAL4}}{TMM_{w_{1118}-GAL4}}\right)$$

For the RNA sequencing, no replicates were available (because these flies were sampled from the remaining individuals left after test day 15), thus no formal statistical test was performed, and altered gene expression level was only reported as expression level relative to the common control line.

### 2.5. Statistical Analysis

The flies’ climbing abilities were analysed using the average distance moved within each vial (i.e., the average of five flies and five pictures), resulting in five observations per knockdown line and the corresponding control line. Using analysis of variance (ANOVA) with F-test it was investigated whether the flies’ climbing ability was reduced with increased age and whether the climbing ability of the RNAi knockdown lines at each age point was significantly reduced compared with the corresponding control line with the same age. All *p*-values were corrected for multiple testing by FDR using a significance level of 0.05. All statistical analyses were conducted in R (version 4.2.1) [37].

## 3. Results

From the four previously published GWAS of sporadic PD [9,12,13,14], we identified 171 genes that harboured genome-wide significant SNPs (i.e., with marginal GWAS *p*-value < $5\times 10^{-5}$), of which 136 were unique genes (Figure 1, Appendix A). Hence, there was a low overlap of the associated PD genes among the four studies (Figure 1).

Among the 136 genes that have been statistically associated with PD, 105 of them had a predicted sequence ortholog in *D. melanogaster* (Figure 1). PD-associated genes that fulfilled the following criteria were selected for functional and behavioural assessment: a DIOPT score > 12, and the upstream activating sequence (*UAS*)-RNAi line available from Vienna *Drosophila* Stock Centre should be homozygous viable. This resulted in a total of 11 genes for subsequent functional validation (Figure 1, Appendix A).

Using the *UAS*-Gal4 system, the gene expression of the 11 selected PD-risk genes was downregulated in adult *D. melanogaster*. The knockdown lines were then phenotypically assessed with the RING assay, and their performance at three time points (age 5, 10, and 15 days) were compared with the common control line (i.e., *w*_1118_-Gal4). After completion of the behavioural assessments (i.e., after age 15 days), the remaining flies were used to determine the level of gene expression reduction by RNA sequencing. As the aim of the study was to determine the phenotypic consequence of gene expression knockdown, we only focused on those RNAi knockdown lines, where we had evidence for some degree of successful gene expression knockdown, thus these results are the first to be described.

Of the eleven selected PD genes that displayed strong evolutionary conservation between humans and *D. melanogaster*, nine of them—based on RNA sequencing experiments—showed successfully ubiquitously downregulated gene expression of the individual target genes compared with their common control line (i.e., the line that had the same genetic background as the knockdown line without the *UAS*-Gal4 insert) (Figure 2, Appendix A).

Flies from the *UAS*-Gal4 lines and the common control line were phenotypically assessed at the age of 5, 10, and 15 days of age (±12 h). The control line did not display any age-related reduced climbing ability (Figure 3), suggesting that natural occurring ageing was not commenced by the age of 15 days. Four of the PD-knockdown lines did display a markedly decreased climbing ability over time (Figure 3, Appendix A), which could suggest accelerated ageing, as decreased climbing ability is a common feature of ageing in *D. melanogaster* [21].

Within each age group, we compared the climbing ability of each knockdown line to their common control line. At the age of 5 days, only flies with gene expression knockdown of *MICU3* displayed a significant 33% reduction in their climbing ability (Figure 4, Appendix A). When flies reached an age of 10 days, five additional lines with reduced gene expression of their target genes (*Ctl1*, *Vha100-1*, *CTsB1*, *CG2066* and *CG33181*) displayed a reduction in their innate escape response, with an observed activity reduction between 17 and 38% (Figure 4, Appendix A) compared with the control line. At the most advanced age-testing time point (day 15), the line with gene expression reduction of *CG7156* also showed a significant 22% reduction in the flies’ climbing ability (Figure 4, Appendix A).

Surprisingly, gene expression knockdown of *HIP1* resulted in a significant 18% increase in activity at age 15 days (Figure 4, Appendix A). Already at age 5 days there was a non-significant indication of increased activity of the *HIP1* knockdown line (Figure 4), which was absent at age 10 days. The significant reduction of 27% in climbing activity for the knockdown of *Vha100-1* at age 10 days disappeared at age 15 days, although the percentage reduction in climbing activity was 38% (Figure 4, Appendix A). The absence of a statistically significant difference between the control line and *Vha100-1* at 15 days is most likely representing a lack of statistical power to detect a difference as many knockdown flies were dead at that time point (Appendix A) and therefore we could not test as many individuals as planned (Appendix A).

## 4. Discussion

During the past decades, technological advancements within molecular biology have advanced our understanding of the genetic architecture of complex human traits including many diseases. Several large-scale efforts, such as the Human Genome Project and the evolution of GWAS, have resulted in a wealth of knowledge about common complex diseases. This includes the discovery of more than 70,000 genetic associations with common diseases and traits [38], which have led to important insights into the underlying genetic aetiology for some diseases. In 2020, the International Common Disease Alliance (ICDA) acknowledged the importance of linking genetic variation to phenotypic variation, but also recognised that it was time to articulate the next phase of common complex trait genetics, namely the paradigm of Maps to Mechanism to Medicine (M2M2M). Central to the M2M2M-framework is the need for functional evidence underlying the genetic associations potentially providing mechanistic knowledge of the aetiology of diseases. In the present study, one step towards the M2M2M-framework was attempted, namely, to systematically investigate genes found to be associated with sporadic PD using *D. melanogaster* as model system.

Here we found considerable evidence that reducing gene expression by RNA-interference of genes previously found to be statistically associated with sporadic PD disturbed the innate climbing ability of the flies. A total of eleven genes were initially included in the study, however, for two of them (*ClC-c* and *EndoA*), no reduction in gene expression level was observed when quantified by RNA sequencing, leaving nine genes left for functional phenotypic investigation. Gene expression knockdown of the PD risk genes resulted in an altered phenotypic behaviour in eight out of nine genetically modified *D. melanogaster* lines. Sporadic PD has a multifactorial aetiology, thus not all knockdown lines were expected to display a phenotypic effect after gene expression knockdown for at least two reasons: (1) the common genetic variants associated with sporadic PD risk only have small effects on the PD risk, and (2) it is likely that some of the selected PD GWAS genes were risk genes associated with non-motor symptoms, which was not captured by the RING assay.

The *MICU3* knockdown line was the only line where reduced gene expression resulted in a consistently altered escape response across all tested age groups (Figure 4). *MICU3* (Mitochondrial calcium uptake 3) is mostly expressed in the adult flies’ neurons and sensory organs [39], where previous findings have indicated that downregulating expression of *MICU3* results in abnormal locomotion [40,41]. In humans, *MICU3* is a brain-specific isoform of the mitochondrial calcium uniporter subunit particularly expressed in astrocytes, thus suggesting that a dysregulated mitochondrial calcium uptake in astrocytes contributes to PD progression [42]. It is not surprising, though, that knockdown of a gene like *MICU3* results in altered behavioural locomotion, as calcium signalling is central for regulating circadian behaviour [43]. *CtsB1* and *CG33181* knockdown lines displayed normal climbing ability at day 5 and exacerbate climbing ability at days 10 and 15 (Figure 3 and Figure 4). Both genes are widely expressed in the adult fly [39], with *CtsB1* encoding a cysteine protease, and *CG33181* encoding a Mg^+^ solute carrier. Recently, knockdown of a magnesium transporter (*Uex*) in a fly model was shown to alter sleep behaviours, likely through a calcium-dependent signalling pathway [44], linking back to the molecular biology of *MICU3*. In humans, the cysteine protease *CTSB*, besides being reported as a potential risk gene for sporadic PD [9,12,13], has also been implicated as a causative factor for the development of Alzheimer’s disease through incomplete proteolytic processing of amyloid precursors [45,46]. This impact of *CTSB* on both Alzheimer’s disease and PD is no surprise as there is growing evidence that the two conditions, i.e., sporadic PD and Alzheimer’s, share common pathological links including genetic risk factors [47,48].

Gene expression knockdown of *Vha1000-1* and *CG7156* reduced the flies’ climbing ability at a single age point (day 10 and day 15, respectively) (Figure 4). In the literature, the ribosomal protein S6 kinase like 1 (*RPS6KL1*) has no clear phenotypic effects, nor has the evolutionary conserved *Drosophila* gene *CG7156*. In *D. melanogaster*, *Vha100-1* is predicted to acidify lysosomes in the neurons, and its ortholog in *H. sapiens*, and the ATPase H+ transporting subunit (*ATP6V01A*) is highly expressed in the human brain [49] and plays an important role in regulation of the pH of lysosomes. Lysosomal dysfunction has been identified as a principal cellular pathology of PD particularly, but also other neurodegenerative diseases such as dementia and amyotrophic lateral sclerosis [50]. Moreover, the human choline transporter, *SLC44A1*, has been suggested to be implicated in the formation of myelin around neurons [51]. Thus, this gene may have important implications for neurodegenerative diseases, such as PD [52], and, as the *D. melanogaster* ortholog is also a choline transporter, this could explain the locomotor deficits observed by gene expression knockdown of *Ctl1* (Figure 4), despite no other mutant phenotypic characteristics previously being described for *Ctl1*. The final line showing deprived climbing ability after gene expression knockdown was *CG32066* (Figure 4). There are no phenotypic records on known consequences of knockdown of gene expression of *CG32066*, however, the human ortholog *FAM49B* is thought to regulate mitochondrial function [53], and mitochondrial dysfunction has been implicated as a potential signature of PD patients [54], and corresponds with the potential evidence from *MICU3*.

Several limitations of our study must be recognised. Firstly, only the flies’ innate escape response, assessed by their climbing ability, was characterised after gene expression knockdown of the selected putative PD risk genes. Thus, only a single axis of the phenotypic spectrum was investigated, and other aspects of PD-like phenotypes, such as other behavioural changes, such as learning or memory deficits, or altered neurological structures, were not investigated. Secondly, to date, 136 genomic loci have been implicated with PD risk, onset, and progression; however, only 8% of these were investigated in the current study. This clearly confines what can be concluded regarding PD risk genes in general. However, under the assumption that gene functionality follows gene conservation, it is pivotal to focus on genes that are evolutionary conserved across species when performing studies like the present one. Of the 136 PD risk loci, 105 genes were found within the *D. melanogaster* genome, with 25 genes displaying strong evolutionary conservatism (Figure 1). Only 11/25 genes were selected for the current study because an emphasis was to obtain RNAi-construct lines that were homozygous viable to ease laboratory labour. Thirdly, genes selected for functional investigation were obtained from large-scale GWAS, where the genetic effect sizes notoriously are moderate to small. Consequently, performing gene expression knockdown of the target genes may not truly recapitulate the underlying biology of the GWAS-associated loci. Finally, *Drosophila* remains purely as a model system where only certain aspects of PD can be investigated. However, the vinegar fly has been used in medical research for decades, because this model species offers experimental possibilities that are absent in other systems that are evolutionarily closer to *H. sapiens*.

In conclusion, we used an unbiased systematic approach to provide further experimental evidence for genes that have been associated with sporadic PD. The aim was not to provide comprehensive molecular characteristics of PD risk genes in *D. melanogaster*, but rather to investigate the possibilities for establishing a framework that feeds into the M2M2M paradigm. The data presented provide considerable evidence that reducing expression levels of PD risk genes in a *D. melanogaster* model system impacts the flies’ innate climbing ability, which suggests that these genes play a role in the dysfunctional locomotion in PD patients.

## Figures and Tables

**Figure 1 insects-14-00168-f001:**
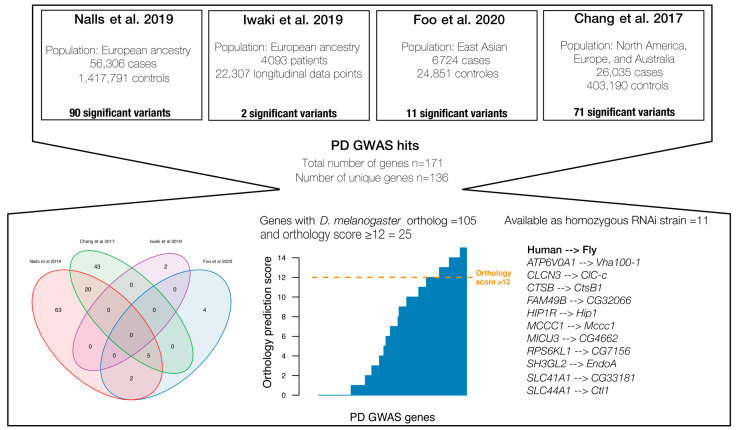
Overview of gene selection from the four large PD genome-wide association studies (GWAS). A total of 136 unique genes were identified, of which 25 had an orthology score above 12. Among the well-conserved PD genes, 11 were available as homozygous viable from Vienna *Drosophila* Stock Centre (VDRC) [33]. Appendix A contains a complete list of all 136 human PD associated genes [9,12,13,14].

**Figure 2 insects-14-00168-f002:**
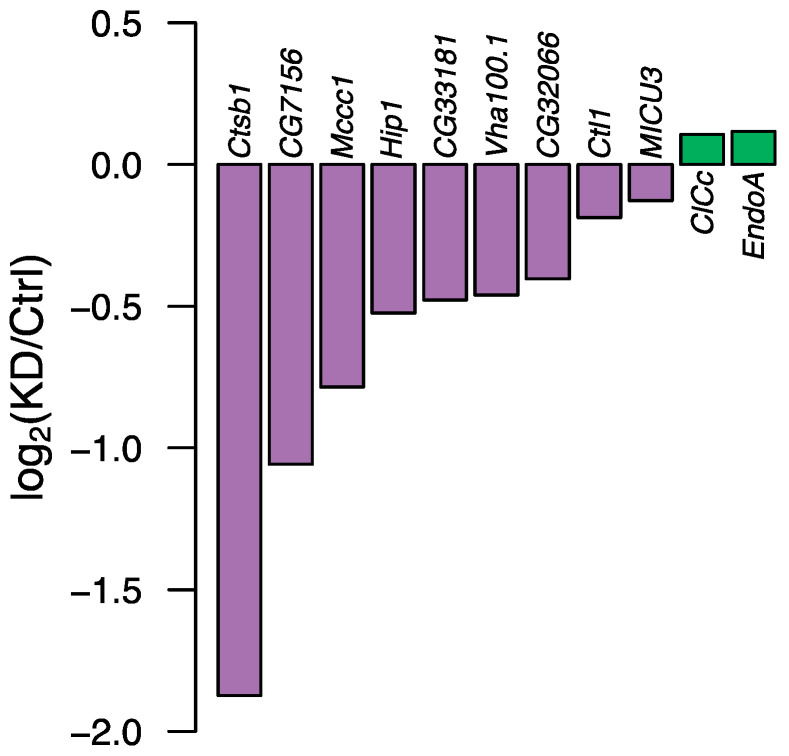
Effect of gene expression knockdown of the 11 selected genes. Each bar represents the log_2_ of the ratio of the TMM normalised gene expression of the target gene for the RNAi line and the control line (*w*_1118_). Purple colours indicate successful reduction in gene expression of the target gene, whereas the green colour indicates upregulation of gene expression. Only knockdown lines with indications of reduced gene expression were used for the phenotypic characterisation. As there are no biological replicates for the RNA sequencing experiments, no error bars can be obtained.

**Figure 3 insects-14-00168-f003:**
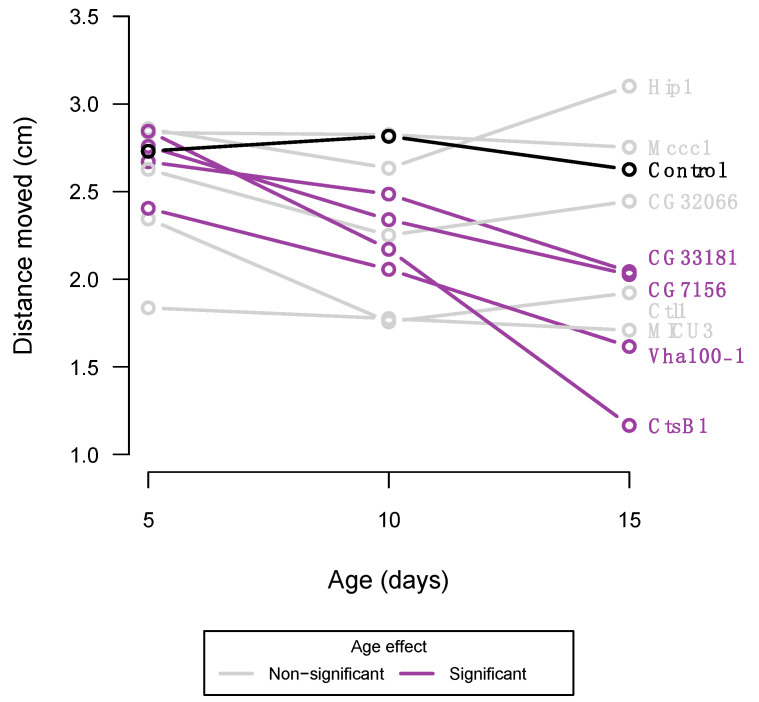
Mean distance climbed per 3 s as function of age for each of the RNAi knockdown lines and the control line (shown in black). Lines highlighted in purple indicate a significant effect of ageing, and their corresponding gene names are listed (together with the control line). Mean values and statistics supporting the figure can be accessed in Appendix A.

**Figure 4 insects-14-00168-f004:**
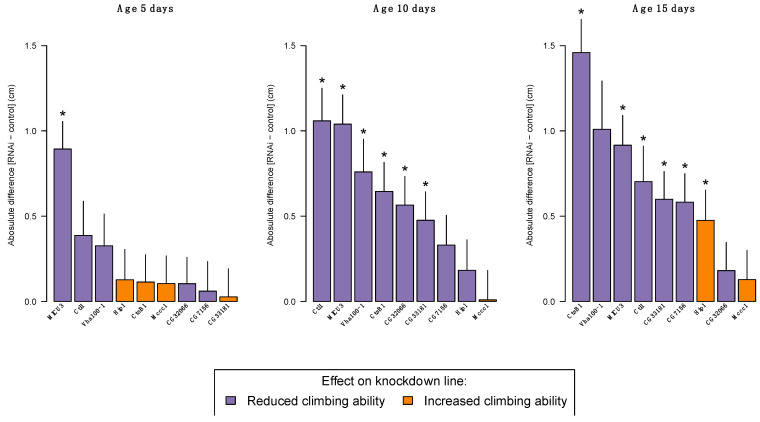
Phenotypic assessment of RNAi lines displaying reduced gene expression of target gene compared with their common control. Each bar represents the absolute difference between a particular RNAi knockdown line and the control line. Error bars represents the standard error of the difference in mean computed as $\sqrt{\frac{SD_{UAS-GAL4}}{n_{UAS-GAL4}}+\frac{SD_{W1118-GAL4}}{n_{W1118-GAL4}}}$. Purple colour indicates reduced climbing ability of the RNAi line, and orange colour indicates increased climbing ability. Asterisks indicate significant age-specific difference between RNAi knockdown and control line (FDR > 0.05). Mean values and statistics supporting the figure can be accessed in Appendix A.

## Data Availability

Available in Appendix A.

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
