# Peer review of "Functionally Validating Evolutionary Conserved Risk Genes for Parkinson’s Disease in Drosophila melanogaster"

_insects, 2023, doi:10.3390/insects14020168_

Round 1
Reviewer 1 Report
insects-2195374
Functionally Validating Evolutionary Conserved Risk Genes for Parkinson’s Disease in Drosophila melanogaster
General comments
In this manuscript, the authors assessed the effect on escape response of flies with down regulation of Drosophila genes whose human orthologs have identified in GWAS studies of Parkinson-disease
It was my pleasure to read this paper, as it overall well written, easy to understand, with a sound approach and appropriate conclusions, as well as discussion of limitations of the study.
Comments:
L36: “The flies’ innate flight response was investigated by assessing their negative geotaxis response”: I do not agree that negative geotaxis assesses the flight response of the flies, rather it assesses their escape response, mostly through their climbing ability. I think that there should not be any mention of flight, only of escape.
L83: 1/ What is the relevance of NUS1? It probably should be added to the lists in the paragraph starting L58.
L83: 2/ “loss of NUS1-expression”, remove the hyphen between loss and expression
L90 – same comment as above about GAK
L141: “5x 15 individuals (10 UAS virgin 141 females and 5 GAL4 virgin males) were separated into five replicates.” Can you please clarify in the text whether you had 5 replicates of 5x15 crosses, or 5 crosses of 1X3 individuals.
L140-147: cross unclear. Based on the text, I understand that UAS virgin females were crossed to Gal4 males, and that from that cross, instead of taking the Gal4;UAS progeny, the following flies were collected and crossed together: +/+;UAS/+ and Gal4/Y;+/+;. Then from that second cross the UAS/+;Gal4/+ male progeny was collected. If my understanding is correct, why performing 2 crosses? Please, explain in the manuscript.
Fig3: indicate the names of all the lines – it is of interest to the reader. I suggest indicating the control in a different color (black?).
Figure 4: wrong number J
L277: “at age 15 // many knock-down flies at that time point were” Do you have data on the survival of those lines?
Table S2 relies on Flybase information, but does not cite it.
Reviewer 2 Report
In this study, the authors use Drosophila as a rapid genetic screening system to investigate how silencing individual genes previously nominated in human sporadic PD GWAS studies affect a PD-related phenotype amenable to testing in flies – innate climbing response in negative geotaxis assays. By doing so, they seek to test whether some of these candidate GWAS genes could contribute to neurodegeneration when perturbed, which would begin to validate their nomination in human GWAS studies.
The use of Drosophila and the RING assay in this regard has merit in that many genes can be screened in a high-throughput manner and at multiple age points due to the relatively short lifespan of flies. The main issue with the current study is that it is very limited in scope. Of 136 genes previously nominated in human GWAS studies, the authors test only 11 in fly studies. There are 105 predicted orthologs in flies of the 136 human genes, but the authors used very strict selection criteria for narrowing down orthologous genes for testing – requiring 12/15 orthology sources from DIOPT in order to be included in the study. This is counterintuitive to the use of Drosophila and fly screens for this kind of study, in that the other ~90 predicted orthologs could have been included (or at least a more inclusive subset of the 105 genes) with the understanding that some of them might not be true orthologs, but in an effort to provide a better estimate of the “hit” rate for phenotypes in flies when studying genes coming from human PD GWAS.
Besides this concern, I suggest one important additional experiment that I believe would be critical to know whether any of the findings from the geotaxis experiment actually relate to the nervous system and therefore PD.
1. Use of the RING climbing assay in this study pertains to PD in that it may uncover potential neurodegenerative phenotypes. However, since this study used ubiquitous RNAi, it’s not clear that the negative geotaxis behavior deficits apparent in many of the RNAi lines tested have anything to do with neurodegeneration – they could be due to deficiencies outside of the nervous system, e.g. in muscle which is important for climbing behavior. Neuronal and or glial (e.g. pan-neuronal, pan-glial or dopaminergic) RNAi follow-up studies should be carried out for each of the candidate genes to distinguish nervous system deficits contributing to lower geotaxis scores from other tissues such as muscle.
Reviewer 3 Report
In the introduction and discussion section, the authors mentioned that this study aims to identify the Parkinson’s disease genes, which are evolutionarily conservative for humans and for Drosophila melanogaster, and to investigate the phenotypical impact of these genes in this model organism. A well-thought-out research concept allowed to clearly demonstrate the relationship between gene function and expression using an excellent research model, which is Drosophila. The experiment was properly designed and conducted. The methodology applied is correctly described and the results are of some interest, but there is an insufficient explanation as to why only males were tested and there are no numbers of individuals that were finally observed in the RING test. The same number of individuals should be tested in all trials, otherwise it is difficult to compare results. This refers especially to Vha100-1 results. The results also do not provide information on fly survival. Despite some minor shortcomings, the article is interesting and deals with an important issue related to PD. The presented findings deliver substantial evidence that downregulation of PD risk genes in the model organism D. melanogaster affects the climbing ability of the fly, suggesting that these genes may play a role in dysfunctional locomotion in PD patients.
Round 2
Reviewer 2 Report
The requested revisions were not performed.